# Hydrochar-Promoted Methane Production in Mesophilic and Thermophilic Anaerobic Digestion of Hydrothermal Pre-Treated Sludge

Chaosen Jing [ID], Chao Zhang, Xingzhang Luo and Zheng Zheng *[ID]

Department of Environmental Science and Engineering, Fudan University, Shanghai 200433, China
* Correspondence: zzhenghj@fudan.edu.cn; Tel.: +86-21-31248911

**Abstract:** Hydrochar produced during the hydrothermal conversion of organic solid waste could enhance the anaerobic digestion (AD) efficiency of hydrothermal pre-treated sludge. However, there was still a lack of systematic research on the effect of hydrochar on improving the methane production and microbial communities of the AD of hydrothermal pre-treated sludge under different temperature conditions. This study explored the effect of hydrochar on methane production from the mesophilic and thermophilic AD of hydrothermal pre-treated sludge and the mechanism of microbial action based on metagenomics analysis. Hydrochar could improve the methane production efficiency of mesophilic and thermophilic AD at different initial concentrations of hydrothermal pre-treated sludge. However, the effect of hydrochar in promoting AD varied under different AD temperatures. Both temperature and hydrochar were crucial factors that could influence the microbial community. Moreover, hydrochar increased the relative abundance of archaea in the AD system, resulting in an increment of 4.99% to 15.30% compared to the control group. Mesophilic reactors exhibit greater microbial diversity. Hydrochar resulted in the significant enrichment of *Synergistota* in the thermophilic AD system and the enrichment of *Firmicutes* in the mesophilic AD system, thereby promoting the hydrolysis of proteins and polysaccharides during AD. This study has practical significance for the resource treatment of excess activated sludge.

**Keywords:** anaerobic digestion; methane production; hydrochar; hydrothermal pre-treated sludge; microbial analysis

## 1. Introduction

Along with the rapid urbanization and industrialization processes, a substantial volume of water resources is being consumed, which might lead to increasing demand for wastewater treatment and a corresponding rise in the annual generation of excess activated sludge [1–4]. Furthermore, the associated management costs related to excess activated sludge disposal impose a considerable economic challenge on wastewater treatment plants (WWTPs), representing over 50% of the complete expenditures on wastewater treatment [5].

Excess activated sludge has a complex composition, containing a series of unutilized organic constituents and substantial amounts of nutrients [6]. Furthermore, it also includes a spectrum of detrimental substances, including heavy metals, persistent organic pollutants, and pathogens [4,5,7,8]. In the absence of timely and appropriate treatment, it has the potential to result in significant secondary pollution. Anaerobic digestion (AD) could effectively transform organic solid waste into $CH_4$ and other biofuels via a sequence of processes, including hydrolysis and acidification, which could not only facilitate the reduction and resource utilization of organic solid waste but also remove most pathogens present in organic solid waste and limit the odor problems related to residual decaying matter [9,10]. Therefore, it was considered a pivotal approach for managing organic solid waste, such as sludge [5]. Nevertheless, the gathering of volatile fatty acids (VFAs) in the AD process could lead to reactor instability and reduce methane production [11,12].

The complicated structure of sludge flocs and the refractory nature of organic constituents within sludge hinder their easy release and subsequent degradation, collectively constraining the efficiency of AD processes [13]. To overcome this constraint, a range of pre-treatment methods, including thermal, mechanical, and biological processes, have been utilized either individually or in combination before AD [14–16]. Hydrothermal carbonization (HTC) is the thermochemical process wherein biomass transforms into solid products (i.e., hydrochar) and liquid products (i.e., bio-oil) under subcritical water conditions, achieved through precise temperature and pressure control (170–350 °C, 4–22 MPa) [17,18]. The hydrochar, as a solid byproduct obtained from HTC, exhibited exceptional hydrophobic characteristics, facilitating its seamless separation from the suspension [19]. Hydrochar has been widely used in environmental remediation, catalyst carriers, and other fields due to its economy and environmental friendliness [20]. Generally, with increased reaction temperature and prolonged reaction time, the hydrochar exhibits a trend of initially increasing and subsequently decreasing oxygen-containing functional groups [19]. The alterations in the properties of hydrochar could correspondingly impact the promoting performance of AD [21]. The pace at which hydrolysis occurs serves as the bottleneck for sludge AD [13]. However, the hydrothermal process, in particular, proves advantageous in liberating a substantial amount of readily biodegradable substances from excess activated sludge, thereby improving the AD efficiency of sludge [22]. The lower hydrothermal temperature and shorter reaction time imply reduced operational costs; hence, it is essential to explore the methane production performance of sludge using hydrothermal treatment under lower pyrolysis temperature conditions [23]. However, excessively low temperatures may compromise the effectiveness of the pre-treatment [24]. Previous studies have revealed a significant enhancement in methane production from residual sludge after treatment at 170 °C for 30 min [21]. Previous studies have also demonstrated that hydrochar generated during the hydrothermal conversion of organic solid wastes could enhance the direct interspecies electron transfer (DIET) process in AD and alleviate the accumulation of VFAs through a mutual metabolism between microorganisms, thereby accelerating methane production [21,25,26]. However, there remains a gap in the impact of hydrochar on the enhancement of methane production and microbial communities in the AD of hydrothermal pre-treated sludge at different AD temperatures. Additionally, previous research has also indicated that AD processes could result in the enrichment of either thermophilic or mesophilic microorganisms within reactors at different temperatures and lead to the formation of distinct microbial community types that exert varying influences on AD [27]. An adequate substrate means more available carbon sources and more $CH_4$ production by microorganisms during AD. However, excessively high substrate concentrations might result in substrate inhibition and lower methane production [28].

Taking into account the aforementioned factors, this study aimed to examine how hydrochar influences the AD process of hydrothermally pre-treated sludge under both mesophilic and thermophilic conditions, including the influence of hydrochar on the efficiency of mesophilic and thermophilic AD in terms of methane production, VFA concentration, and pH. Furthermore, metagenomic analysis was also used to elucidate the enhancement mechanisms of AD in relation to microbial communities.

## 2. Materials and Methods

### 2.1. Inoculum and Sludge

The inoculum employed in this research was obtained from an upflow anaerobic sludge blanket (UASB) reactor. The inoculum underwent a 5 min grinding process in a grinder (QX-650, Qinxin, China) to achieve homogenization prior to the experiment. The inoculated sludge exhibited the following physical and chemical properties: volatile solids (VS) $30.1 \pm 1.9$ g/L, total solids (TS) $37.2 \pm 2.8$ g/L, and pH $7.2 \pm 0.1$.

The preparation process of hydrothermal pre-treated sludge used in this study was as follows: Sludge samples were collected from the secondary sedimentation tank at a WWTP in Shanghai. Detailed information about the WWTP can be found in Text S1.

The samples were placed in a centrifuge (TG16-WS, Cence, Changsha, China) in batches at a speed of 9500 rpm/min to separate water and sludge until their moisture content reached approximately 90%. Subsequently, 6 L sludge samples were placed in a 10 L pressurized stirred-tank reactor (JWOGF-H10L, Taikang, Xi'an, China) with agitation for preparation. The stirring rate was set at 200 rpm, with a heating rate of 1 °C/min, and the reaction temperature was maintained at 170 °C. After a 30 min reaction period at 170 °C, cooling was initiated by introducing condensed water. Once the internal temperature of the reaction vessel decreased to room temperature, the hydrothermal sludge was retrieved for subsequent batch experiments. The physicochemical properties of the hydrothermal pre-treated sludge were as follows: TS 31.75 ± 0.8 g/L, VS 25.54 ± 0.7 g/L, and pH 6.9 ± 0.1.

*2.2. Hydrochar Production*

Hydrochar was prepared using cornstalk (moisture content 21.4%) as the raw material. The collected cornstalk was sieved through a 40-mesh sieve after being pulverized. The resultant mixture of cornstalk and deionized water, with a mass ratio of 1:10, was introduced into a 1 L pressurized stirred-tank reactor (EasyChem, Century SenLong experimental, Beijing, China). Referring to the previous study by Shi et al., the reactor temperature was increased to 260 °C with a rate of 10 °C/min, and the reaction was carried out at 260 °C and 5 MPa pressure for 1 h [21]. The entire reaction process involved stirring at a rate of 200 rpm. After the reaction, solid and liquid phases were separated using a suction filter bottle with a 0.45 μm filter membrane and a circulating water vacuum pump (SHZ-D(III), Huachen Instruments, Shanghai, China). To prevent the influence of bio-oil on the solid surface of the AD process, multiple washes with tetrahydrofuran (THF) were conducted as a precautionary measure. Specifically, the hydrothermal charcoal was soaked in tetrahydrofuran (10 mL THF/g hydrochar), sonicated for 30 min, and centrifuged to discard the THF. This process was repeated several times until the THF was clear. Given the toxicity of THF to microorganisms, 95% ethanol was employed for a secondary rinsing of residual tetrahydrofuran on the hydrothermal charcoal surface, continuing until the ethanol solution was clarified after washing. Subsequently, the hydrochar was dried in a 60 °C oven for 24 h and ground to pass through a 100-mesh sieve for further use.

*2.3. Batch Experiments*

Batch experiments were conducted using 118 mL serum bottles equipped with butyl rubber stoppers. The operational volume within each serum bottle was set at 40 mL, which included hydrothermally treated sludge, deionized water, and inoculated sludge. The specific dosage of the hydrothermal treatment sludge, deionized water, and inoculated sludge in each reactor is shown in Table S1. The concentration gradient of hydrothermally treated sludge in the serum bottles was set at 5.0 g VS/L, 10.0 g VS/L, and 20.0 g VS/L. In addition, 10 g/L of hydrochar was introduced into the serum bottles, and a control group without hydrochar was established. Blank groups without hydrothermally treated sludge were established to monitor gas production from the inoculated sludge. Pre-experiments confirmed that the hydrochar used in this study did not produce methane. The starting pH in each serum bottle was set to 7.5 using HCl or NaOH. Nitrogen gas was purged for 5 min, and then the bottles were sealed with butyl rubber stoppers to maintain anaerobic conditions. The sealed bottles were placed in temperature-controlled incubators at 35 °C and 55 °C, respectively. Six parallel bottles were set up for each experimental condition, with three bottles used for measuring methane production and the other three for assessing parameters such as pH, VFAs, and microbial composition.

The experimental groups were named based on the following format: "Temperature (M, mesophilic; H, thermophilic)—Presence of Hydrochar (C, no hydrochar; H, hydrochar added)—Initial Substrate Concentration (L, initial concentration of 5.0 g VS/L; M, initial concentration of 10.0 g VS/L; H, initial concentration of 20.0 g VS/L)". For example, the conditions for the MC-M group reactor were 35 °C with an initial sludge concentration of 10.0 g VS/L and no hydrochar added. After the experiments, samples from four groups,

MC-M, MH-M, HC-MH, and HH-M, were selected for metagenomic analysis based on their methane production performance.

### 2.4. DNA Extraction and Sequencing

The total genomic DNA was collected using the PowerMax Soil DNA Isolation Kit (MoBio Laboratories, Carlsbad, CA, USA) [29]. The quality of the genomic DNA was assessed using Nanodrop ND-2000c (Thermo Fisher Scientific, Waltham, MA, USA). The construction of metagenomic libraries was performed using the Illumina Nextera DNA Library Preparation Kit [22]. DNA sequencing was carried out on the Illumina HiSeq 2000 platform, generating 150 base pair paired-end reads. Raw sequencing data were uploaded to the China Bioinformation Center with project number PRJCA021203.

### 2.5. Metagenomic Analysis Methods

The FastQC software was employed to trim the paired-end reads [30]. MEGAHIT (v1.0) was utilized to assemble the metagenome [31]. Metagenome binning was conducted based on the default parameters of MetaBAT2, CONCOCT, and MaxBin2. Subsequently, the bins produced using these three binning software were amalgamated using the MetaWRAP pipeline and subjected to redundancy removal [32]. The integrity and contamination levels of microbial genomes (MAGs) were determined using CheckM (v1.0.4) [33], which evaluates them by identifying and quantifying single-copy marker genes. Only those genomes estimated to have a completeness exceeding 90% and contamination levels below 5% were retained. The MAGs taxonomic classification was executed using gtdb-tk (v0.1.3) with the "classify_wf" function [34].

### 2.6. Analytical Methods

VFA concentrations in the collected liquid samples from the anaerobic reaction process were analyzed using HPLC (Agilent Technologies, Santa Clara, CA, USA) according to the method of He et al. [26]. The TS and VS were determined following the procedures outlined in the Standard Methods [35].

Data analysis was performed using SPSS and Excel, and the corresponding charts were generated using OriginPro 2019b. A Spearman correlation analysis and one-way ANOVA were conducted using SPSS Statistics v20. A principal component analysis (PCA) was conducted using the vegan package in R 4.1.0. Microbial abundance heatmaps were generated using the heatmap package in R 4.1.0.

## 3. Results

### 3.1. Hydrochar Enhanced the Mesophilic AD of Hydrothermal Pre-Treated Sludge

The effect of hydrochar on the mesophilic AD process under different substrate concentrations was explored by controlling the initial concentration of hydrothermal pre-treated sludge. As the concentration of hydrothermal pre-treated sludge increased, the cumulative methane production in the MC-L, MC-M, and MC-H reactors gradually decreased in the absence of hydrochar (Figure 1), which might be attributed to the accumulation of VFAs resulting from the hydrolysis and fermentation of high-concentration organics within the reactors, which subsequently inhibited the anaerobic reaction process. However, with the addition of hydrochar, there was an overall improvement in AD efficiency across reactors with different sludge concentrations, which suggests that hydrochar could effectively enhance the AD process of hydrothermal pre-treated sludge. At an initial hydrothermal pre-treated sludge concentration of 5.0 g VS/L, the MC-L group achieved an actual methane production of $190.66 \pm 11.15$ mL $CH_4$/g VS on the 30th day, while the MH-L group reached a stable methane production by the 24th day, with an actual methane production of $200.00 \pm 3.75$ mL $CH_4$/g VS on the 30th day. With an elevation in the initial concentration of hydrothermally pre-treated sludge, the impact of hydrochar addition on augmenting the actual methane production in reactors showed improvement. Furthermore, we observed that as the initial hydrothermal pre-treated sludge concentration increased, the effect of hydrochar addition

on enhancing actual methane production in reactors improved. At an initial hydrothermal pre-treated sludge concentration of 10.0 g VS/L, the actual methane production of the MC-M group without hydrochar added on the 30th day was 157.43 ± 4.22 mL $CH_4$/g VS, while the MH-M group with hydrochar addition achieved an actual methane production of 218.88 ± 6.77 mL $CH_4$/g VS, resulting in a 39.03% increase compared to the group without hydrochar. When the initial concentration of hydrothermal pre-treated sludge was 20.0 g VS/L, on the 30th day, the actual methane production of the MH-H group with hydrochar added was 149.80 ± 4.92 mL $CH_4$/g VS, which showed a significant increase of 339.10% compared to MC-H. It could be seen that hydrochar significantly enhanced methane production, especially under lower substrate concentrations.

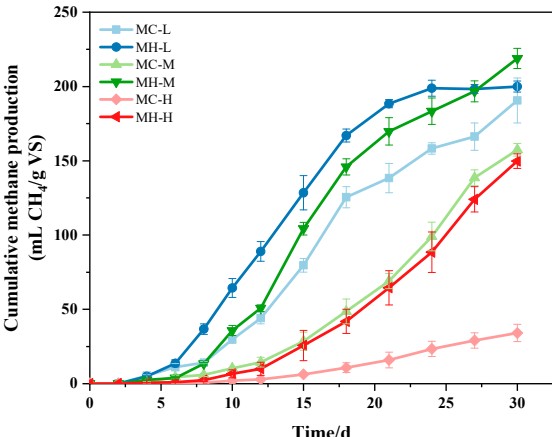

**Figure 1.** Influence of hydrochar on cumulative methane production of hydrothermal pre-treated sludge with different concentrations during mesophilic AD.

The methane production during the AD process for each group was modeled using the modified Gompertz model (Text S2), and the corresponding kinetic parameters for the methane production process are presented in Table S2. The maximum methane production rates ($R_m$) for the MC-L, MC-M, and MC-H groups were 11.29 ± 0.70 mL/d, 9.23 ± 0.49 mL/d, and 2.07 ± 0.03 mL/d, respectively. This indicated that the $R_m$ of the reactor without hydrochar addition gradually decreased with an increasing initial hydrothermal pre-treated sludge-addition concentration. In addition, the $R_m$ for the MH-L, MH-M, and MH-H groups were 15.25 ± 0.86 mL/d, 14.70 ± 0.75 mL/d, and 9.93 ± 0.35 mL/d, respectively. It indicated that as the initial hydrothermal pre-treated sludge-addition concentration increased, all reactors with hydrochar exhibited a gradually decreasing trend in the $R_m$ of the hydrothermal pre-treated sludge AD process. Moreover, there are notable distinctions in $R_m$ between the reactors with hydrochar addition and those without. Compared to the MC-L, MC-M, and MC-H groups, the $R_m$ in the MH-L, MH-M, and MH-H groups with hydrochar increased by 35.05%, 59.26%, and 380.13% ($p < 0.05$). This indicated that hydrochar could not only enhance methane production during hydrothermal pre-treated sludge AD but also improve the $R_m$ of hydrothermal pre-treated sludge.

Apart from $R_m$, the lag time $\lambda$ was also a crucial indicator to evaluate the efficiency of the hydrothermal pre-treated sludge AD process. The increase in $\lambda$ with the elevated initial concentration of hydrothermally pre-treated sludge may be attributed to the generation of toxic compounds during the hydrothermal carbonization process [36]. The $\lambda$ of the MC-L, MC-M, and MC-H groups were 7.71 ± 0.46, 12.68 ± 0.51, and 12.99 ± 0.20 days, respectively. When the initial hydrothermal pre-treated sludge concentrations were 5.0 g VS/L and 10.0 g VS/L, the addition of hydrochar could significantly shorten the $\lambda$ of the AD process ($p < 0.05$), with a $\lambda$ of 6.10 ± 0.35 and 8.09 ± 0.33 days for the MH-L and MH-M groups, respectively.

However, under the conditions of an initial hydrothermal pre-treated sludge concentration of 20.0 g VS/L, hydrochar increased the $\lambda$ in the hydrothermal pre-treated sludge AD process. The $\lambda$ for the MC-H and MH-H groups were 12.99 ± 0.20 and 14.82 ± 0.45 days,

respectively. It was noted that the $R_m$ of the MH-H group surpassed that of the MC-H group. This might be attributed to the gradual consumption of VFAs in the MH-H group as the reaction progressed, while the MC-H group remained restricted by hydrolysis. To investigate the reasons further and understand the influence of hydrochar in the hydrothermal pre-treated sludge AD process, the changes in pH and VFAs were also measured. Acetate, propionate, and butyrate emerged as the primary VFA types in hydrothermal pre-treated sludge AD reactions. In the initial 4–8 days of hydrothermal pre-treated sludge AD, the concentration of VFAs in the reactor with an initial hydrothermal pre-treated sludge concentration of 5.0 g VS/L gradually increased (Figure 2a). The cumulative concentration of VFAs reached the maximum on the eighth day, and the cumulative concentrations of VFAs in the MC-L and MH-L groups reached 914.68 mg/L and 1202.96 mg/L, respectively. After the 12th day of AD, the VFAs in the reactor were rapidly consumed. Adding hydrochar could enhance the conversion and consumption of VFAs in the AD process of hydrothermal pre-treated sludge, thereby reducing the accumulation of VFAs in the AD reactor. It is worth noting that reactors with hydrochar addition exhibited higher VFA accumulation concentrations compared to those without hydrochar on the fourth and eighth days. This observation was likely attributed to the function of hydrochar in promoting the breakdown of organic matter in the substrate, which facilitated the acidification step of AD [21]. Reactors with initial hydrothermal pre-treated sludge concentrations of 10.0 g VS/L and 20.0 g VS/L showed a similar trend in the total VFAs concentration as the reactor with 5.0 g VS/L. VFAs initially accumulated and then decreased over time in all reactors. By the eighth day of the experiment, the reduction in VFA concentrations was more pronounced in the hydrochar-added groups compared to the control group without hydrochar, and this trend became more significant with the increasing initial hydrothermal pre-treated sludge concentration. At the end of the experiment, reactors with the lowest initial concentrations of hydrothermal pre-treated sludge (MC-L and MH-L) exhibited VFA accumulations of 21.26 mg/L and 7.50 mg/L, respectively (Figure 2b). In contrast, reactors with the highest initial hydrothermal pre-treated sludge concentrations (MC-H and MH-H) had VFA accumulations of 1835.28 mg/L and 321.59 mg/L on the 31st day, which corresponded to the higher $R_m$ in the hydrochar-added groups (Figure 2c). In experiments with an initial hydrothermal pre-treated sludge concentration of 5.0 g VS/L, both the experimental and control groups contained a substantial microbial population capable of promptly metabolizing the organic compounds and intermediate products within the hydrothermal pre-treated sludge. These microorganisms could efficiently consume the VFAs generated during the anaerobic process, thereby mitigating the immediate promoting effect of hydrochar addition. The results indicated that hydrochar exhibited a more pronounced capacity to facilitate VFA utilization and degradation in anaerobic reactors operating under high organic load conditions. In summary, this study demonstrated that hydrochar had an impact on VFA concentrations during the AD of hydrothermal pre-treated sludge. Hydrochar could not only promote the hydrolysis and acidification steps in the AD process of hydrothermal pre-treated sludge, improving the degradation rates of soluble carbohydrates and soluble proteins but also enhance the consumption of VFAs and other organic substances in reactors operating under high organic loads.

The fluctuations in pH during the AD of hydrothermal pre-treated sludge were closely associated with the existence of VFAs. The production and accumulation of VFAs on the fourth and eighth days resulted in a pronounced decrease in pH across all reactors. Specifically, on the eighth day, the pH for the MC-L, MC-M, and MC-H groups decreased to 7.17, 6.98, and 6.7, respectively (Figure 2d). Notably, the MH-L, MH-M, and MH-H groups exhibited even lower pH values (7.13, 6.89, and 6.63, respectively) than their corresponding control groups, aligning with the respective VFA concentrations within the reactors. And by the 12th day, the rapid pH increase indicated the extensive consumption of VFAs. The pH within all groups remained relatively stable after 24 days, signifying the near-complete depletion of accumulated VFAs.

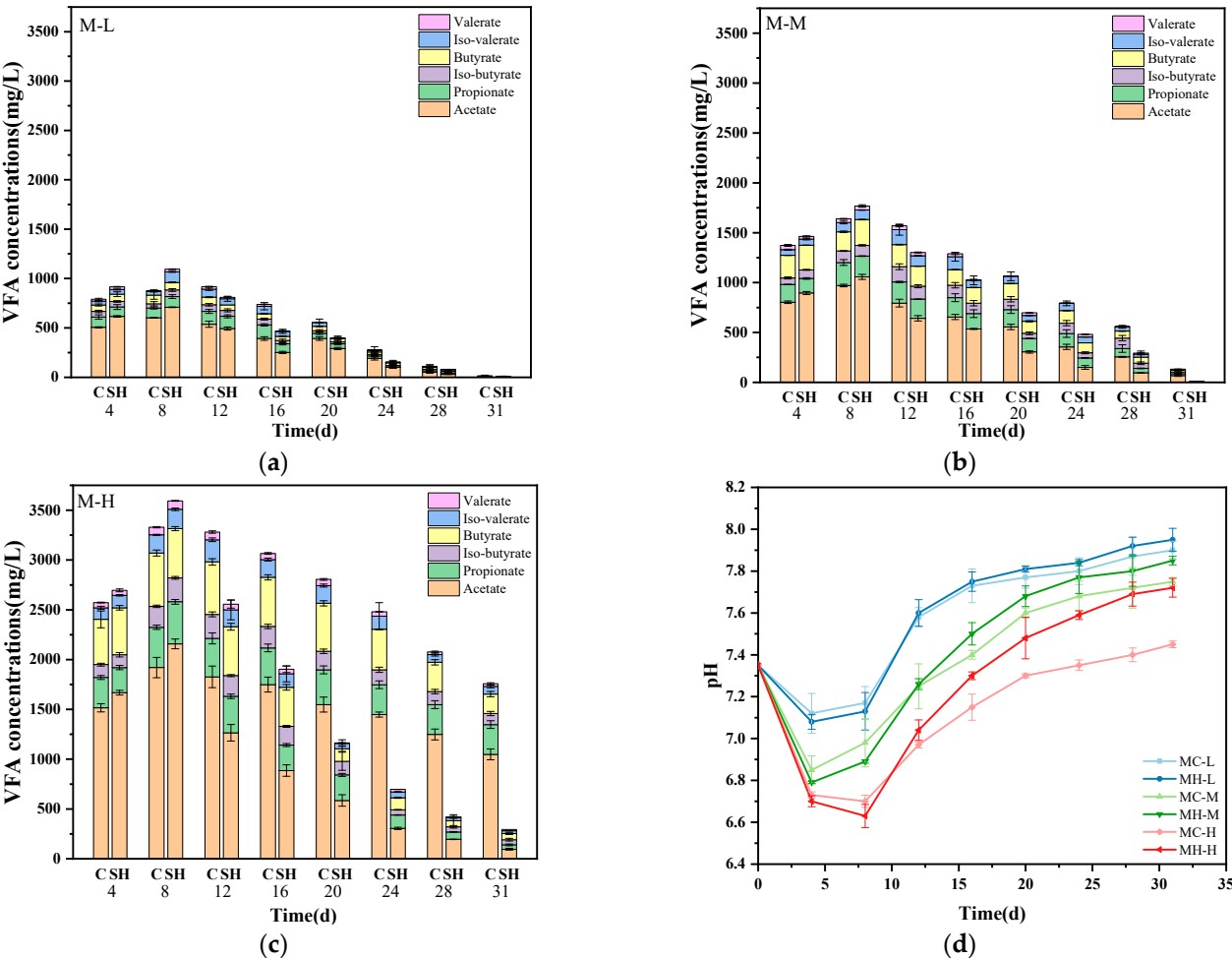

**Figure 2.** Effect of hydrochar on VFAs variation (**a**–**c**) and pH variation (**d**) of hydrothermal pre-treated sludge with different concentrations during mesophilic AD. In panels (**a**–**c**), the notation "C" designates the group without hydrochar, while "SH" designates the group with hydrochar.

### 3.2. Hydrochar Enhanced the Thermophilic AD of Hydrothermal Pre-Treated Sludge

The methane production in the thermophilic AD of hydrothermal pre-treated sludge at different concentrations is illustrated in Figure 3. On the 30th day of AD, the HH-L group achieved an actual total methane production of $209.17 \pm 6.76$ mL $CH_4$/g VS, representing a 28.66% increase compared to the HC-L group. Under conditions of an initial hydrothermal pre-treated sludge concentration of 10.0 g VS/L, the HH-M group reached an actual total methane production of $197.01 \pm 10.56$ mL $CH_4$/g VS, a substantial increase of 118.13% over the methane production without hydrochar addition. The MH-H group exhibited an actual total methane production of $27.20 \pm 1.40$ mL $CH_4$/g VS, marking a remarkable enhancement of 130.89% compared to the HC-H group. Overall, the actual methane production during the AD of hydrothermal pre-treated sludge gradually increased with increasing sludge concentration. Moreover, at low, medium, and high hydrothermal pre-treated sludge concentrations, the AD efficiency within the reactors showed improvement, which suggested that hydrochar could effectively promote the thermophilic AD of hydrothermal pre-treated sludge.

The relevant kinetic parameters of the AD process are shown in Table S3. With the increase in the initial addition concentration of hydrothermal pre-treated sludge, the $R_m$ of the AD process exhibited a gradual decrease. For reactors without hydrochar addition, at an initial hydrothermal pre-treated sludge concentration of 5.0 g VS/L, the $R_m$ was $8.46 \pm 0.28$ mL/d. When the initial sludge concentration was raised to 10.0 g VS/L, the $R_m$ decreased to $5.76 \pm 0.32$ mL/d. As the initial sludge concentration further increased

to 20.0 g VS/L, the $R_m$ dropped to $0.86 \pm 0.04$ mL/d. The addition of hydrochar in the HH-L, HH-M, and HH-H groups increased the $R_m$ by 96.65%, 92.78%, and 104.94%, respectively, compared to the corresponding reactors without hydrochar addition at the initial hydrothermal pre-treated sludge concentrations ($p < 0.05$). Under thermophilic AD conditions, there was a significant difference in the $R_m$ between reactors with the hydrochar addition and those without, indicating that hydrochar could effectively enhance $R_m$ in the thermophilic AD of hydrothermal pre-treated sludge.

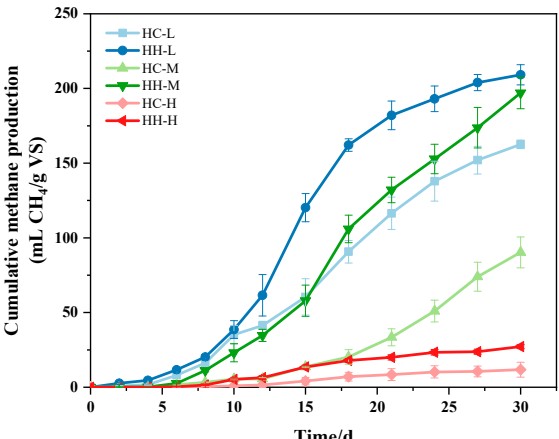

**Figure 3.** Influence of hydrochar on cumulative methane production of hydrothermal pre-treated sludge with different concentrations during thermophilic AD.

Furthermore, under conditions where the initial hydrothermal pre-treated sludge concentration was 10.0 g VS/L and 20.0 g VS/L, the addition of hydrochar significantly reduced the λ of AD. For reactors without hydrochar addition (HC-L group, HC-M group, and HC-H group), the λ were $7.18 \pm 0.30$, $14.54 \pm 0.53$, and $10.06 \pm 0.30$ days, respectively. The higher λ in the HC-H group might be influenced by the microbial processes occurring within the reactor. In the cases of moderate and high initial sludge concentrations, the addition of hydrochar significantly reduced the λ of AD ($p < 0.05$). When the initial hydrothermal pre-treated sludge concentration was 10.0 g VS/L and 20.0 g VS/L, the λ in the HH-M and HH-H groups were reduced to $9.13 \pm 0.33$ and $7.77 \pm 0.45$ days, corresponding to a decrease of 37.19% and 22.78%, respectively. However, in reactors with low initial sludge concentration, the HH-L group showed slightly higher λ compared to the control group. Nevertheless, the HH-L group exhibited significantly higher actual methane production and $R_m$ than the control group, indicating that hydrochar still enhanced methane production in low initial sludge conditions.

Subsequently, we continued to investigate changes in pH and VFAs during the thermophilic AD of hydrothermal pre-treated sludge. During the initial 4–8 days of AD, the VFA concentrations gradually increased in all reactors. By the 8th to 12th day of the experiment, the VFA accumulation reached its peak. In reactors with an initial hydrothermal pre-treated sludge concentration of 20.34 g VS/L, the HC-H and HH-H groups exhibited the highest VFAs accumulation, reaching 3230.53 mg/L and 3004.2 mg/L, respectively (Figure 4c). Under initial hydrothermal pre-treated sludge concentrations of 5.0 g VS/L and 10.0 g VS/L, the highest VFAs accumulation during AD was observed, reaching 999.08 mg/L (HC-M group on the 12th day) and 1721.51 mg/L (HC-H group on the 12th day). As VFAs accumulated in the reactors, the pH rapidly decreased, and this decrease was more pronounced in reactors with higher initial hydrothermal pre-treated sludge concentrations. In reactors without hydrochar addition, the pH reached its lowest point on the 12th day of the experiment. The pH for the HC-L, HC-M, and HC-H groups on the 12th day were 7.10, 6.96, and 6.89, respectively. On the eighth day, the pH reached its lowest point in the HH-L, HH-M, and HH-H groups, with values of 7.11, 7.00, and 6.90, respectively (Figure 4d). Reactors with hydrochar addition reached their lowest pH earlier

than those without hydrochar, and the pH remained more stable during AD compared to the reactors without hydrochar. This observation could be associated with dominant functional microorganisms in the system and variations in microbial substrate metabolic pathways [21,29]. From the 12th to the 16th day of the experiment, the VFAs in the reactors were rapidly consumed and converted into $CH_4$ and $CO_2$, leading to a gradual increase in pH. The maximum methane production was also observed during this phase. After the 24th day, VFA accumulation decreased, and the consumption rate gradually slowed down, resulting in pH stabilization. This pattern aligned with the gradual reduction in methane production in the system. At the end of the experiment, the pH for different initial hydrothermal pre-treated sludge concentrations (5.0 g VS/L, 10.0 g VS/L, and 20.0 g VS/L) stabilized within the ranges of 7.89–7.94, 7.71–7.85, and 7.55–7.59, respectively.

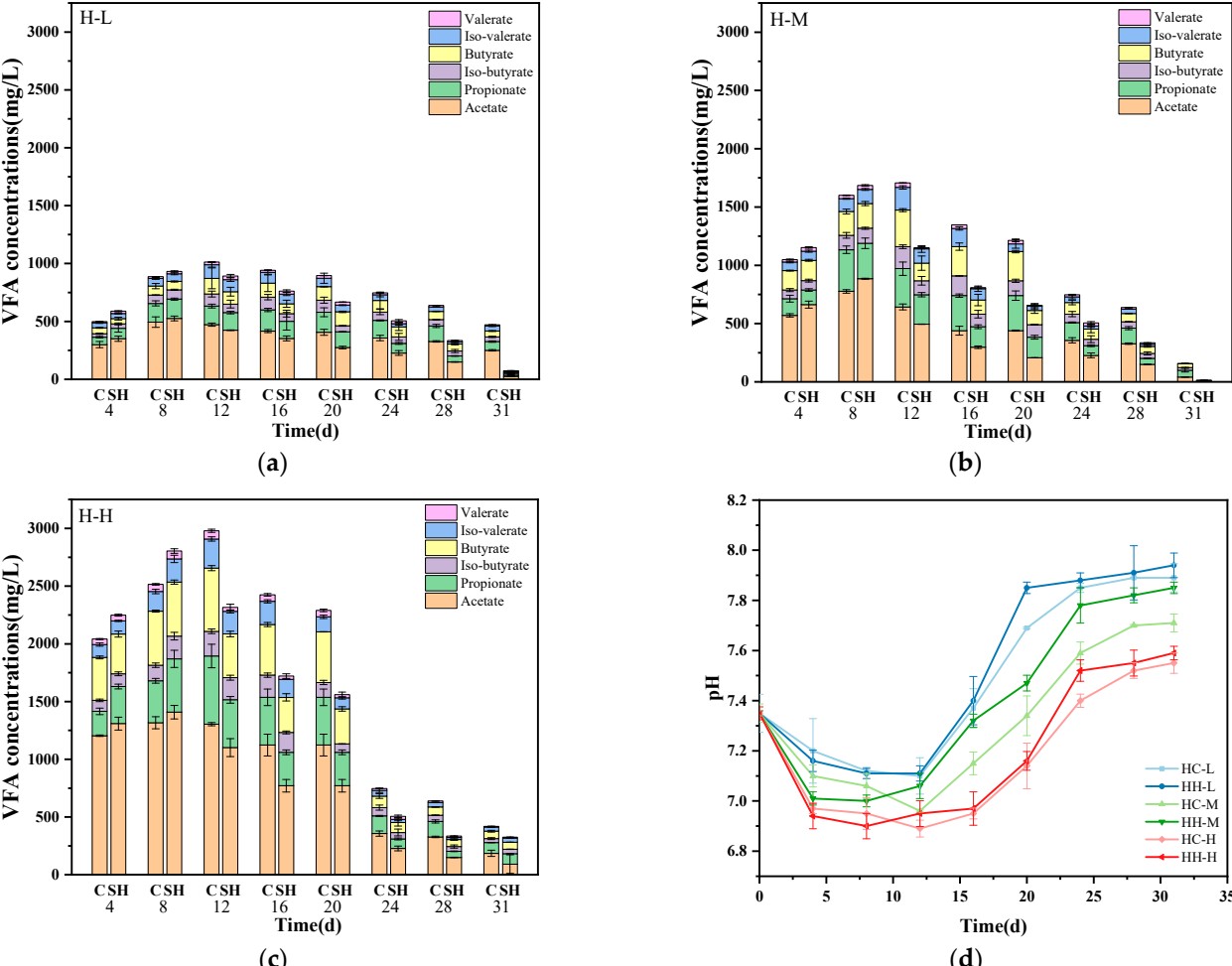

**Figure 4.** Effect of hydrochar on VFAs variation (**a**–**c**) and pH variation (**d**) of hydrothermal pre-treated sludge with different concentrations during thermophilic AD. In panels (**a**–**c**), the notation "C" designates the group without hydrochar, while "SH" designates the group with hydrochar.

### 3.3. Effect of Temperature on the AD of Hydrothermal Pre-Treated Sludge

Previous studies have shown that temperature was closely associated with AD efficiency, with the suitable temperature range playing a critical role in the process [37,38]. Furthermore, temperature could significantly affect methane production by influencing the stability of enzymes and coenzymes [38,39]. Anaerobic microorganisms could thrive under low temperature (10–30 °C), mesophilic (30–40 °C), or thermophilic (50–60 °C) conditions. However, since methane production is relatively low at low temperatures, mesophilic and thermophilic AD was usually used in practical applications [37,40]. Therefore, this study focused on the AD of hydrothermal pre-treated sludge at 35 °C and 55 °C. At the end of the ex-

periment, a significant contrast in methane production was evident between the mesophilic and thermophilic AD of hydrothermal pre-treated sludge. Under the same initial hydrothermal pre-treated sludge concentrations, the mesophilic reactors exhibited higher cumulative methane production compared to their thermophilic counterparts (Figures 1 and 2). The most significant difference in cumulative methane production between the mesophilic and thermophilic AD was observed at an initial hydrothermal pre-treated sludge loading of 10.0 g VS/L. At the end of the experiment, the MC-M group showed an increase of 66.11 mL $CH_4$/g VS in cumulative methane production compared to the HC-M group. In contrast, the MC-L and MC-H groups had an increase of 22.33–28.09 mL $CH_4$/g VS in cumulative methane production over their corresponding thermophilic reactors at similar loading concentrations. When the hydrothermal pre-treated sludge loading was below 10.0 g VS/L, the enhancement in $R_m$ via hydrochar addition was more pronounced in thermophilic anaerobic reactors. However, when the hydrothermal pre-treated sludge loading reached 20.0 g VS/L, the enhancement effect of mesophilic AD was significantly better than that of thermophilic AD. At lower hydrothermal pre-treated sludge concentrations, the hydrochar addition had a more significant impact on maximum methane production in thermophilic reactors than in mesophilic reactors. When the hydrothermal pre-treated sludge concentration exceeded 20.34 g VS/L, the improvement effect of hydrochar in mesophilic AD was more prominent.

*3.4. Effect of Hydrochar on Microbial Community Diversity*

Metagenomic binning methods could isolate individual sequences (reads or contigs) from complex microbial communities, making a strain-level functional analysis based on individual genome assembly possible [41,42]. Considering the significant methane enhancement effect of hydrochar addition under the initial hydrothermal pre-treated sludge concentration of 10.0 g VS/L, after the experiment, the MC-M group under mesophilic conditions (abbreviated as MC1, MC2, and MC3), the MH-M group (abbreviated as MH1, MH2, and MH3), the HC-M group under thermophilic conditions (abbreviated as HC1, HC2, and HC3), and the HH-M group (abbreviated as HH1, HH2, and HH3) were selected to explore the influence of hydrochar and temperature on the microbial community in those reactors. Sludge samples were collected from the reactors for metagenomic sequencing analysis. Metagenomic sequencing produced 193.4 GB of paired-end data after deduplication and quality filtering. Finally, 144 medium-high quality (completeness >70% and contamination <5%) metagenomic assembled genome MAGs were reconstructed from the collected sludge samples for subsequent analysis (Figure 5). Phylogenetic analysis showed that these 144 MAGs were taxonomically divided into 18 bacterial phyla and 3 archaeal phyla.

Hydrochar significantly influenced the biodiversity during the AD of hydrothermal pre-treated sludge (Figure 6a). The Shannon index serves as a measure of biodiversity or species diversity within a given ecological community, with higher values indicating greater diversity and lower values suggesting lower diversity in the community [29]. The Shannon index in the four groups of reactors follows this order: MH > HH > MC > HC. The Shannon index was higher in the mesophilic AD microbial community, indicating a more diverse and richer ecosystem in the mesophilic AD system, with a greater variety of organisms involved in the hydrothermal pre-treated sludge AD process. On the other hand, the addition of hydrochar resulted in a higher Shannon index compared to the control group at the corresponding temperature, which showed that hydrochar could improve hydrothermal pre-treated sludge by increasing alpha diversity in both mesophilic and thermophilic AD systems.

A PCA analysis based on Bray–Curtis distance reveals a distinct separation in the distribution patterns of the microbial community composition among the MC, MH, HC, and HH groups (Figure S1). MC were notably separated from the other groups, which revealed that both temperature and hydrochar played substantial roles in influencing the microbial community. The interaction between various microorganisms could be promoted by regulating the anaerobic reaction temperature or adding hydrochar. And the functional

microorganisms in the system could establish connections with each other more effectively, thereby promoting the methanogenesis process of hydrothermal pre-treated sludge. The hierarchical clustering analysis of the microbial community similarity levels among various anaerobic reactors and the construction of a dendrogram further confirm the differences in microbial community composition among the reactors. These findings collectively illustrated that both anaerobic reaction temperature and the addition of hydrochar could lead to distinct microbial community structures within the reactors.

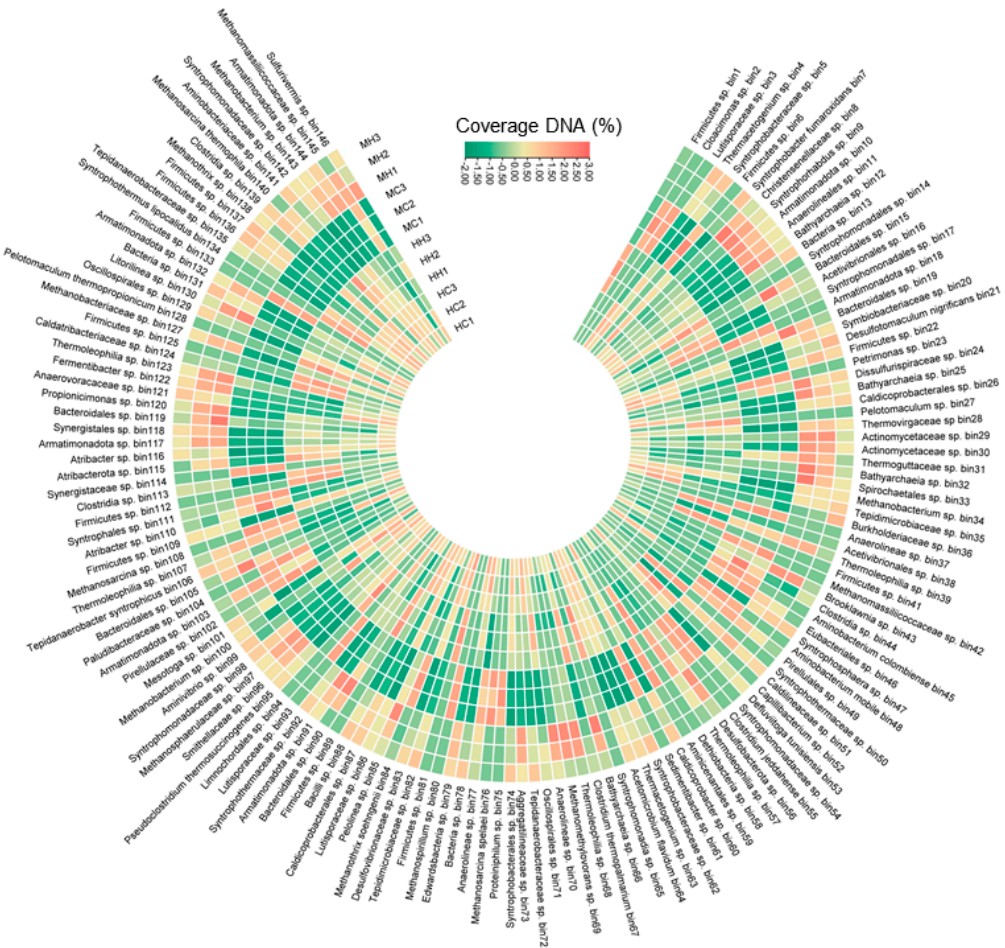

**Figure 5.** Abundance of reconstructed bacterial and archaeal MAGs in each reactor.

A total of 66.67% of the MAGs were shared among all the reactors (Figure 6b). The MC group possessed the highest number of unique MAGs, which suggested that reaction temperature was a critical factor influencing the microbial community. Additionally, the MH group also exhibited a significant number of unique MAGs, which indicated that hydrochar might result in the emergence of more distinct MAGs in the mesophilic AD reactor. Despite the presence of numerous distinctive microorganisms in the HC group, its methane production rate was lower than that in the HH group. This discrepancy might be attributed to competitive interactions between the unique microorganisms enriched in the HC group and the functional bacteria involved in the methane production process, which, in turn, reduce the microbial abundance and richness participating in methane production. The presence of distinctive microbial taxa in the MH group might also be associated with α-diversity, as they surpass those in the MC group. This might be due to differing mechanisms by which hydrochar promotes methane production under mesophilic and thermophilic conditions. In conclusion, these findings revealed that both temperature and hydrohar could lead to the formation of distinct microbial communities, consequently impacting the AD of hydrothermal pre-treated sludge.

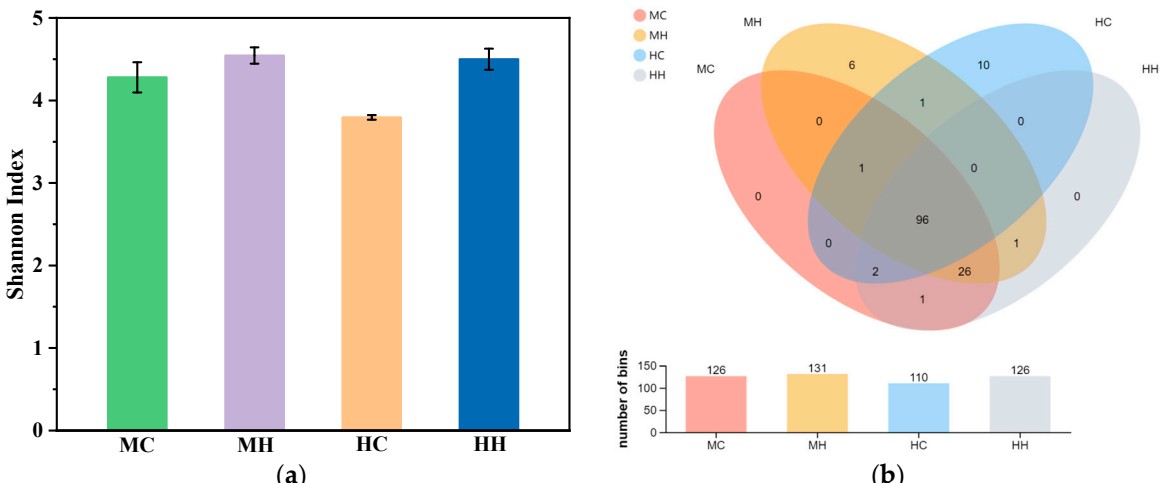

**Figure 6.** Microbial community diversity in mesophilic and thermophilic AD. (**a**) Microbial community Shannon index. (**b**) Venn diagram of common and unique bins in microbial communities in mesophilic and thermophilic AD systems.

### 3.5. Microbial Community Composition and Dynamics

To gain a more comprehensive understanding of how hydrochar influences the composition of the microbial community in the hydrothermal pre-treated sludge AD system, this study further analyzed the distribution of bacteria and archaea in each reactor. The relative abundance of archaea in the reactor ranged from 59.9% to 70.83% in HH and HC. The abundance of archaea in thermophilic AD reactors was generally lower than that in mesophilic AD reactors (Figure 7a). In addition, hydrochar resulted in an increased abundance of archaea in the reactors, which corresponded to the high $R_m$ of each group with hydrochar addition. In the HH group, the archaeal relative abundance was 62.4%, exhibiting a 4.99% increase compared to the HC group. Additionally, the MH group displayed an archaeal relative abundance of 70.83%, marking a 15.30% rise compared to the HC group. This phenomenon suggested that hydrochar was more conducive to enriching archaea in mesophilic AD.

Subsequently, the distribution of microorganisms in each reactor was analyzed at the gate level for a comprehensive insight into the influence of hydrochar on the hydrothermal pre-treated sludge AD process. The results showed that *Methanobacteriota*, *Halobacterota*, *Firmicutes*, *Actinobacteriota*, and *Synergistota* were the dominant microbial phyla in all reactors (Figure 7b). Among them, *Methanobacteriota* was the most abundant microbial phylum in all reactors, accounting for 33.84%, 37.43%, 40.94%, and 42.16% of all microbial abundance in the MC group, MH group, HC group, and HH group, respectively. It is worth noting that *Methanobacteriota* is one of the essential functional bacteria in the methane process. Some bacteria belonging to *Methanobacteriota* could use acetate, $H_2/CO_2$, and methanol for methane production during AD. In addition, there were some bacteria, such as *Syntrophobacter* and *Hydrogenogens*, that could engage in cooperative hydrogenotrophic methane production processes. Hydrochar addition also resulted in a notable rise in the relative abundance of *Synergistota* and *Firmicutes* in the MH and HH groups compared to the MC and HC groups without hydrochar addition. Some microorganisms belonging to *Synergistota* possess the capability to degrade peptides, amino acids, and proteins, the augmented relative abundance of which plays a role in converting complex organic matter in hydrothermal pre-treated sludge into small-molecule substances, thereby promoting methane production in subsequent processes [29]. *Firmicutes* have also been documented to be the prevailing microorganisms in AD reactors [43]. *Firmicutes* include members with various functions, including polysaccharide and protein hydrolysis, glycolysis, and acidogenesis [22]. Furthermore, the addition of hydrochar resulted in a noteworthy decline in the relative abundance of *Halobacterota* in mesophilic and thermophilic AD reactors.

While bacteria belonging to the *Halobacterota* exhibit strong adaptability, high salt tolerance, and various special functions in high-salinity environments, they are typically not the primary participants in AD processes. In summary, the addition of hydrochar to mesophilic and thermophilic AD reactors for hydrothermal pre-treated sludge enables the control of the microbial community, thereby promoting the AD process.

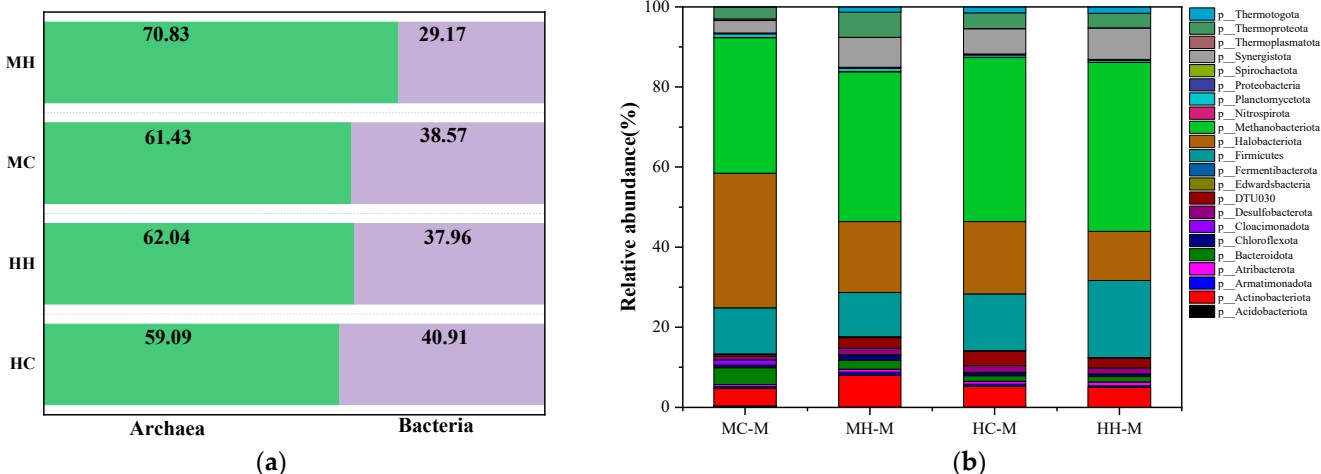

**Figure 7.** Microbial community structure in mesophilic and thermophilic AD. (**a**) Abundance percentage of bacteria and archaea. (**b**) Phylum-level microbial community structure.

## 4. Discussion

This investigation reveals notable variations in methane production between mesophilic and thermophilic AD of hydrothermal pre-treated sludge. The actual methane production in the mesophilic groups surpasses that in the corresponding reactors under thermophilic conditions. However, in the initial phases of the AD process, the thermophilic AD reactors exhibit higher methane production than the mesophilic ones. Previous research has indicated that the thermophilic AD process holds an advantage over the mesophilic AD process as it enhances the breakdown of volatile solids during the initial hydrolysis phase of anaerobic reactions. In contrast to thermophilic AD, mesophilic AD could provide more stable methane production.

The contribution of hydrochar to the enhancement of methane production was evident in two aspects. Firstly, hydrochar could promote the hydrolysis and acidification steps of hydrothermal pre-treated sludge and mitigate the excessive accumulation of VFAs. Previous studies also indicated that hydrochar addition might promote the degradation of soluble carbohydrates and soluble proteins during AD [22]. Secondly, hydrochar could also have an impact on the microbial community; the addition of hydrochar-enriched *Methanosarcina* and *Acetomicrobium*. The enhancement of *Acetomicrobium*, a common microorganism in thermophilic sludge that could convert glucose into acetate, $CO_2$, and $H_2$, might explain the effect of hydrochar on thermophilic AD [44]. It has been reported that *Methanosarcina* could accept electrons to reduce $CO_2$ and produce methane through DIET. DIET usually relies on cytochrome c and e-pili, enabling the direct transfer of electrons between contacting methanogens and bacteria [45]. Hydrochar could help some microorganisms that cannot form conductive flagella complete the DIET process by acting as a conductor between microorganisms. Conversely, symbiotic bacteria could leverage the surface of hydrochar for electron conduction. It was also found that there was a positive correlation between the presence of oxygen-containing functional groups on the hydrochar surface and the efficiency of AD [46]. Moreover, the adsorption capabilities of hydrochar also played a role in capturing toxic substances within the reactor, mitigating their harmful effects on microorganisms and consequently impacting methane generation [47].

This study was concerned with the impact of hydrochar on enhancing the AD performance of hydrothermal pre-treated sludge. The sludge contains a high density of

microorganisms and various pollutants. Therefore, the influence of hydrochar on their distribution should also be taken into consideration. Moreover, it is worth noting that the effective utilization and treatment of digestate and biogas slurry generated during anaerobic digestion is also a significant consideration. In the future, there is an urgent need for research investigating the resource utilization of digestate and biogas slurry in anaerobic digestion reactors supplemented with hydrochar.

## 5. Conclusions

Hydrochar positively enhanced methane production efficiency in mesophilic and thermophilic AD systems with different initial hydrothermal pre-treated sludge concentrations. However, the influence of hydrochar on AD efficiency varies depending on the digestion temperature. At initial hydrothermal pre-treated sludge concentrations below 10.0 g VS/L, hydrochar significantly improved the $R_m$ in thermophilic AD reactors. And when the initial hydrothermal pre-treated sludge concentration reaches 20.0 g VS/L, the enhancement in mesophilic AD efficiency via hydrochar becomes more pronounced than in the thermophilic condition. In addition, both temperature and the addition of hydrochar were crucial factors that could influence the microbial community. Hydrochar elevated the relative abundance of archaea in the AD system, resulting in an increment of 4.99% to 15.30% compared to the control group. Mesophilic reactors exhibit greater microbial diversity.

**Supplementary Materials:** The following supporting information can be downloaded at https://www.mdpi.com/article/10.3390/fermentation10010010/s1, Figure S1: comparative plot of composition PCA analysis of microbial community structure under different conditions based on Bray–Curtis distance; Table S1: The specific dosage of hydrothermal treatment sludge, deionized water, inoculated sludge, and hydrochar in each AD reactor; Table S2: modified Gompertz equation fitting methanogenic kinetic parameters of hydrothermal sludge with different initial concentrations during mesophilic AD; Table S3: modified Gompertz equation fitting methanogenic kinetic parameters of hydrothermal sludge with different initial concentrations during thermophilic AD; Text S1. The modified Gompertz model. Text S2. Detailed information about wastewater treatment plants.

**Author Contributions:** Conceptualization, C.J. and Z.Z.; methodology, C.J.; software, C.Z.; validation, C.J. and X.L.; formal analysis, C.J.; investigation, C.J.; resources, Z.Z.; data curation, C.J.; writing—original draft preparation, C.J.; writing—review and editing, C.Z.; visualization, C.J.; supervision, X.L.; project administration, Z.Z.; funding acquisition, X.L. and Z.Z. All authors have read and agreed to the published version of the manuscript.

**Funding:** This research was funded by ABA Chemicals and the National Major Science and Technology Special Project for Water Pollution Control and Treatment (2017ZX07602-001-005).

**Institutional Review Board Statement:** Not applicable.

**Informed Consent Statement:** Not applicable.

**Data Availability Statement:** Raw sequencing data were uploaded to the China Bioinformation Center with project number PRJCA021203.

**Acknowledgments:** This study was financially supported by ABA Chemicals and the National Major Science and Technology Special Project for Water Pollution Control and Treatment (2017ZX07602-001-005). We thank our colleagues and students from Fudan University for their assistance with the measurements.

**Conflicts of Interest:** The authors declare no conflicts of interest. The funders had no role in the design of the study, in the collection, analyses, or interpretation of data, in the writing of the manuscript, or in the decision to publish the results.

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
