# Peer review of "Hydrochar-Promoted Methane Production in Mesophilic and Thermophilic Anaerobic Digestion of Hydrothermal Pre-Treated Sludge"

_fermentation, doi:10.3390/fermentation10010010_

Round 1
Reviewer 1 Report
Comments and Suggestions for Authors
Please see the attached file.

Author Response
Thank you very much for taking the time to review this manuscript. we have carefully revised the manuscript based on your comments. Please see the attachment.

Reviewer 2 Report
Comments and Suggestions for Authors
The authors set the stage to explore the effect of hydrochar on methane production from mesophilic and thermophilic AD of hydrothermal pretreated sludge and the mechanism of microbial action based on metagenomics analysis. The results revealed that at hydrothermal pretreated sludge concentrations below 10 g VS/L, hydrochar significantly improved Rm in thermophilic AD reactors. When the concentration reaches at 20.0 g VS/L, the improvement in mesophilic AD efficiency by hydrochar becomes more pronounced than in the thermophilic condition. Both temperature and hydrochar were crucial factors that could influence the microbial community. Moreover, hydrochar increased the relative abundance of archaea in AD system, resulting in an increment of 4.99% to 15.30% compared to the control group. This study seems interesting and i suggest it for publication
Author Response
Thank you very much for taking the time to review this manuscript.
Round 2
Reviewer 1 Report
Comments and Suggestions for Authors
line 52 – please replace “was” with ‘is’
line 53-54 – “transformation into liquid products and bio-oil” please add “solid”
Line 54-58 – “The resulting solid byproduct, hydrochar, was seamlessly separated from the suspension owing to its remarkable hydrophobic characteristics[19]. The properties of hydrochar were influenced by various factors, including reaction temperature and reaction time[20,21]” It is written as methodology not as a review. Please rephrase it.
Regarding author response no. 2 – The provided information still does not support the process conditions studied in this work. Why these specific conditions (170 °C and 30 minutes for sewage sludge and 260 °C and 60 min for cornstalk) were chosen? Was it chosen randomly? Please provide an explanation in the introduction or in the methodology section supporting this choice.
Author Response
Thank you very much for taking the time to review this manuscript. we have carefully revised the manuscript based on your comments.
Comments 1: line 52 – please replace “was” with ‘is’
Response 1: We are very sorry for our incorrect writing and related sentence was corrected as seen in Line 53 “Hydrothermal carbonization (HTC) is the thermochemical process…..”
Comments 2: line 53-54 – “transformation into liquid products and bio-oil” please add “solid”
Response 2: Thanks for the comments. We have added related information as seen in Line 87-88 “Hydrothermal carbonization (HTC) is the thermochemical process wherein biomass un-dergoes transformation into solid products (i.e., hydrochar) and liquid products (i.e., bio-oil) under subcritical water conditions, achieved through precise temperature and pressure control (170-350°C, 4-22MPa)[17,18].”.
Comments 3: Line 54-58 – “The resulting solid byproduct, hydrochar, was seamlessly separated from the suspension owing to its remarkable hydrophobic characteristics[19]. The properties of hydrochar were influenced by various factors, including reaction temperature and reaction time[20,21]” It is written as methodology not as a review. Please rephrase it.
Response 3: Thanks for the comments. We have rewritten the sentence as seen in Line 52-59 “The hydrochar, as solid byproduct obtained from HTC, exhibited exceptional hydrophobic characteristics, facilitating its seamless separation from the suspension[19]. Hydrochar has been widely used in environmental remediation, catalyst carrier and other fields due to its economy and environmental friendliness[20]”.
Comments 4: Regarding author response no. 2 – The provided information still does not support the process conditions studied in this work. Why these specific conditions (170 °C and 30 minutes for sewage sludge and 260 °C and 60 min for cornstalk) were chosen? Was it chosen randomly? Please provide an explanation in the introduction or in the methodology section supporting this choice.
Response 4: Thanks for the comments. We have added related information as seen in Line 66-72 “The lower hydrothermal temperature and shorter reaction time imply reduced operational costs; hence, it is essential to explore the methane production performance of sludge through hydrothermal treatment under lower pyrolysis temperature conditions[24]. How-ever, excessively low temperatures may compromise the effectiveness of the pretreatment[25]. Previous study have revealed a significant enhancement in methane production from re-sidual sludge after treatment at 170°C for 30 minutes[22]” and Line 116-118 “Referring to the previous study by Shi et al., the reactor temperature was increased to 260 °C with a rate of 10 °C/min, and the reaction was carried out at 260 °C and 5 MPa pressure for 1 hour[22]”.